# Cysteine-Mediated Green Synthesis of Copper Sulphide Nanoparticles: Biocompatibility Studies and Characterization as Counter Electrodes

**DOI:** 10.3390/nano12183194

**Published:** 2022-09-14

**Authors:** Luis A. Saona, Jessica L. Campo-Giraldo, Giovanna Anziani-Ostuni, Nicolás Órdenes-Aenishanslins, Felipe A. Venegas, María F. Giordana, Carlos Díaz, Mauricio Isaacs, Denisse Bravo, José M. Pérez-Donoso

**Affiliations:** 1BioNanotechnology and Microbiology Lab, Center for Bioinformatics and Integrative Biology (CBIB), Facultad de Ciencias de la Vida, Universidad Andres Bello, Av. República #330, Santiago 8370186, Chile; 2Instituto de Física Rosario (IFIR-CONICET-UNR), Bvrd. 27 de Febrero 210 Bis, Rosario 2000, Argentina; 3Instituto de Ciencias, Universidad de Las Américas, Sede Providencia, Manuel Montt 948, Santiago 7500975, Chile; 4Departamento de Química Inorgánica, Facultad de Química, UC Energy Research Center and Centro de Investigación en Nanotecnología y Materiales Avanzados, Pontificia Universidad Católica de Chile, Santiago 7820436, Chile; 5Laboratorio de Microbiología Oral, Facultad de Odontología, Universidad de Chile, Sergio Livingstone PohlHammer #943, Santiago 8380453, Chile

**Keywords:** fluorescent nanoparticles, Cu_2_S NPs, counter electrode, green synthesis

## Abstract

A one-pot green method for aqueous synthesis of fluorescent copper sulphide nanoparticles (NPs) was developed. The reaction was carried out in borax–citrate buffer at physiological pH, 37 °C, aerobic conditions and using Cu (II) and the biological thiol cysteine. NPs exhibit green fluorescence with a peak at 520 nm when excited at 410 nm and an absorbance peak at 410 nm. A size between 8–12 nm was determined by dynamic light scattering and transmission electron microscopy. An interplanar atomic distance of (3.5 ± 0.1) Å and a hexagonal chalcocite crystalline structure (βCh) of Cu_2_S NPs were also determined (HR-TEM). Furthermore, FTIR analyses revealed a Cu-S bond and the presence of organic molecules on NPs. Regarding toxicity, fluorescent Cu_2_S NPs display high biocompatibility when tested in cell lines and bacterial strains. Electrocatalytic activity of Cu_2_S NPs as counter electrodes was evaluated, and the best value of charge transfer resistance (R_ct_) was obtained with FTO/Cu_2_S (four layers). Consequently, the performance of biomimetic Cu_2_S NPs as counter electrodes in photovoltaic devices constructed using different sensitizers (ruthenium dye or CdTe NPs) and electrolytes (S^2−^/S_n_^2−^ or I^−^/I^3−^) was successfully checked. Altogether, novel characteristics of copper sulfide NPs such as green, simple, and inexpensive production, spectroscopic properties, high biocompatibility, and particularly their electrochemical performance, validate its use in different biotechnological applications.

## 1. Introduction

Metal-based semiconductor nanocrystals of elements such as Ag, Zn, Cd, Se and Cu, among others, have become increasingly relevant given their unique size-dependent optical, physicochemical and electronic properties [1,2]. Among these, metal sulphide nanoparticles (NPs) have been widely studied in basic and applied science due to their multiple applications.

To date, there are many methods to synthesize Cu_x_S_x_ NPs. These include microemulsion [3], sol–gel and different metal reduction processes such as electrochemical, hydrothermal, thermal [4], and sonochemical reductions [5]. However, most of these methods display limitations related to the use of toxic solvents, hazardous chemical precursors, inert atmospheres, high temperatures, and high pressures. All these conditions render NPs with different features, and most of them have poor water solubility and biocompatibility [6].

Green synthesis approaches involve the use of biological molecules or conditions to synthesize biocompatible and soluble NPs. Peptides, biopolymers, proteins, amino acids, and particularly biological thiols are currently used in the green synthesis of metal NPs [7]. In general, the use of biomolecules in NPs-synthesis involves several advantages: (i) biomolecules can act as scaffold or template for NPs nucleation and growth [8]; (ii) soluble biomolecules participate in NPs size control [9]; (iii) biomolecules can act during the synthesis as reducing agent [10]; and (iv) biomolecules can also be used as stabilizers or capping agents [11].

Most copper nanoparticles produced to date have commercial importance and are constituted by Cu^0^, CuO, and Cu_x_S_y_. Nevertheless, in recent years copper sulphide NPs have attracted great attention because of their wide range of stoichiometric compositions [12,13], and their application as pigments, catalysts in lithium secondary batteries, semiconductors, and superconductors in optoelectronics devices [14,15]. Furthermore, the application of Cu_x_S_y_ NPs as eco-friendly and efficient materials to fabricate Quantum Dots Sensitized Solar Cells (QDSSCs) has gained importance during the last decade [16]. QDSSCs have become more attractive due to the global need of replacing fossil fuels as an energy source and considering that the sun is the more abundant energy source on Earth [17].

Like dye-sensitized solar cells (DSSCs), QDSSCs are mainly constituted by three parts: the electron acceptor, the electrolyte, and the counter electrode (CE). CE is responsible for the electron transfer from the external circuit to the redox electrolyte and mediates the reduction reaction of species in the electrolyte [18]. To date, the most common CE used in solar cells is the rare and expensive material, Pt. However, there is growing interest in developing new and cheaper materials to be used as CE [19]. Previous results indicated that Cu-sulphide nanomaterials constitute a real alternative to replace Pt as CE in solar cells [20]. In this context, the use of Cu in photovoltaic research involves several benefits when compared with other metals; it is more abundant, less expensive, and highly biocompatible to human cells [21].

Solving energy requirements is extremely important for developing countries that economically depend on commodities. This is especially relevant for mining countries such as Chile, which besides being the top Cu-producing nation, receives high levels of solar radiation all year round. In this context, the generation of Cu-based photovoltaic technologies that give “added value” to Cu and that also contribute to energy requirements, constitutes a unique opportunity for countries such as Chile, Argentina, Bolivia, and Peru, among many others.

Despite the large number of Cu_x_S_y_ NP synthesis processes reported to date [12,13,14,15,16,17], the remaining challenge is a green chemical method that synthesizes safe NPs with physicochemical properties that allow them to be used in a wide range of applications. In this regard, here we report a new eco-friendly, safe, and economical method to synthesize Cu_2_S NPs; the characterization of these NPs reveals that they possess improved properties. The NPs showed aqueous solubility, low toxicity, and organic coating. Finally, the application of the produced NPs as a counter electrode was demonstrated, which is a crucial first step for the adoption of this new technology by reducing the cost of solar cells fabrication by avoiding the use of Pt.

## 2. Materials and Methods

### 2.1. Synthesis of Cu_2_S Nanoparticles

The synthesis of fluorescent copper nanoparticles was carried out in aqueous media using copper chloride (Sigma-Aldrich, Burlington, MA, USA). CuCl_2_ 0.3 mM, L-cysteine 14 mM (Sigma-Aldrich) and borax–citrate buffer pH 9.4 were used for the synthesis. The solutions were incubated at 37 °C with constant stirring for 12 h. The color of the solution changed from colorless to golden brown, phenomenon that was indicative of NPs formation. The samples were dialyzed using dialysis tubing cellulose membranes to remove remaining salts and reagents. Then, the dialyzed samples were centrifuged on ultra-4 Millipore centrifugal filters to collect nanometer-sized products. Finally, samples were lyophilized on a SP Scientific Virtis Benchtop Sentry 2.0 Freeze Dryer to obtain a golden powder for further characterization.

### 2.2. NPs Characterization

The absorption and emission spectra of NPs were recorded using a Synergy H1M multiple-well plate reader (Biotek, Winooski, VT, USA). FTIR analyses were obtained by using a Spectrum Two (Perkin Elmer, Waltham, MA, USA), between 4000 and 400 cm^−1^. The size of nanoparticles was measured through dynamic light scattering on a Zetasizer Nano ZS. HR-TEM micrographs of NPs were obtained with a TEM FEI Tecnai F20 G2 with field emission gun operated at 200 kV. The sample was suspended in phosphate buffered saline and dispersed ultrasonically to separate individual particles. Then, the suspension was deposited onto a 300-mesh holey carbon copper grid (Electron Microscopy Sciences).

### 2.3. NPs Toxicity in Eukaryotic and Prokaryotic Cells

To assess the cytotoxicity of NPs on eukaryotic cells, an MTS ([3-(4,5-dimethylthiazol-2-il)-5-(3-carboxymethoxyphenyl)-2-(4-sulfophenyl)-2H-tetrazolium) assay was performed in TERT2 OKF6-type eukaryotic cells. A total of 20,000 cells per well were incubated for 24 h at 37 °C and 5% CO_2_ to allow adherence of cells to the plate. After 24 h, cells were treated with different concentrations and dilutions of Cu_2_S NPs. Again, treated cells were incubated for 24 h at 37 °C and 5% CO_2_. Then, cells were washed with PBS (3 times) and treated with 100 mL of a mixture of phenazine methosulphate and MTS; on the ratios indicated by the manufacturer (Promega). After 1 h of incubation, the absorbance at 490 nm was measured.

Toxicity of NPs to prokaryotic cells was evaluated in a Gram-positive (*S. aureus*) and Gram-negative (*E. coli*) bacterium. Both bacteria were grown in LB broth in Erlenmeyer flasks at 37 °C to an optical density at 600 nm of 0.3 and supplemented with different concentrations of Cu_2_S NPs at that time. NPs concentrations used were 100, 500, 1000, and 2000 μg/mL. To determine the effect of NPs on the growth of microorganisms, absorbance was measured at 600 nm at different times until reaching the stationary phase.

### 2.4. Fabrication of Quantum Dot Sensitized Solar Cells

QDSSCs were produced following the protocol described by Órdenes-Aenishanslins et al., (2014) with some modifications. To fabricate the electrodes of QDSSCs, 10 × 10 × 2 mm size fluorine doped tin oxide coated glasses (FTO) TEC15, with a surface resistivity of 13 [Ω/sq] and 85% transmittance were used. Conductive glasses were cleaned by successive sonication in absolute ethanol and deionized water for 10 min to remove organic contaminants. The Cu_2_S counter electrode was prepared by adding 100 μL of Cu_2_S NPs on ultrapure water solution on FTO coated glass and heated 20 min at 350 °C. This step was repeated several times, cleaning the surface with deionized water each time. The anode was prepared using a suspension of titanium (IV) oxide nanoparticles (TiO_2_ nanopowder, ~21 nm particle size and anatase crystal structure, Sigma-Aldrich Co.) that was deposited on the glass through spin coating at 2000 rpm for 10 sec. The electrodes (TiO_2_ films) underwent a sintering process at 465 °C for 20 min, creating an active area of 1.0 cm^2^. Sensitization of TiO_2_ film was performed by direct adsorption of CdTe quantum dots [2]. Then, the photoanode and the counter electrode were assembled leaving a 127-μm space between them. Before sealing the cell, 7 μL of electrolyte solution was added. Sulphide/polysulphide (S^2−^/S_n_^2−^) (NaOH 0.1 M, Na_2_S 1.0 M, and S 0.1 M on ultrapure water) or iodide/triiodide (I^−^/I_3_^−^) (Iodolyte AN-50, Solaronix S.A., Aubonne, Switzerland) redox mediators were used.

### 2.5. Electrode Impedance Spectroscopy

All electrochemical experiments were measured on a CH Instrument model 760C electrochemical workstation. EIS experiments were performed using sandwiched cells (E/electrolyte/CE) with a Kapton tape as spacer and a S^2−^/S_n_^2−^ redox couple electrolyte. Nyquist plots were acquired between 10^−5^–10^−1^ Hz, in dark conditions and the initial potential was set at the open circuit potential (OCP) measured previously for every cell over 400 s. Data were fitted with Z view software.

### 2.6. Characterization of Quantum Dot Sensitized Solar Cells

Characterization of solar cells were performed with a solar simulator (A1 Solar LightLine, Sciencetech Inc., London, ON, Canada) and a current-voltage measurement system (IV Tester- 20W, Sciencetech Inc.). Measurements were performed under constant conditions of temperature and irradiance at a one sun intensity as the light source (~100 mW·cm^−2^ and AM1.5).

## 3. Results and Discussion

### 3.1. Synthesis and Characterization of Cu_2_S NPs

Copper NPs were synthesized by incubating a Cu-precursor and cysteine in borax–citrate buffer (pH 9.4) for 12 h with constant stirring (aerobic conditions). When cysteine and Cu were mixed in the buffer, the solution immediately turned a brown color. Then, the solution color moved to golden indicating the generation of Cu_x_S_y_ NPs (see methods for details).

Optical properties of nanomaterials strongly influence their final applications. Therefore, absorbance and fluorescence properties of copper NPs produced by this method were investigated. Biomimetic NPs show a maximum absorption peak at 410 nm, in agreement with previous reports showing absorption peaks between 350 and 410 nm for Cu NPs [3]. The shoulder at 410 nm corresponds to the chalcocite phase (Cu_2_S). The lack of absorbance near the IR region suggests that Cu NPs do not present the covellite phase (CuS) [4,5,16].

Synthesized Cu-NPs emit green fluorescence when exposed to UV light through a transilluminator (ʎ_ext_ = 360 nm) (data not shown). Accordingly, the fluorescence spectrum shows a strong and defined emission peak with maximum emission at 520 nm when excited at 410 nm (Figure 1). A fluorescence emission profile near 520 nm for Cu_x_S_y_ NPs has been previously reported [5]. Nevertheless, fluorescence was obtained when NPs were excited at 250 nm and copper NPs display broad emission peaks.

To determine organic molecules and potential Cu_x_S_y_ bonds present in the metal nanostructure, biomimetic NPs and a cysteine standard were analyzed by FTIR. In both IR spectra, a band between 1570 and 1690 cm^−1^ corresponding to the vibration of the carbonyl group (C=O) is observed (Figure 2). For carbonyl groups it is common to observe a peak near 1700 cm^−1^, however, for both spectra, this peak is shifted to shorter wavelengths than it have been previously described on L-cysteine-capped CdTe NP [22]). This shift can be a consequence of the formation of a cysteine amino acid zwitterion [23]. In this case, the carboxylate group presents a resonance effect in which both C-O bonds share the electron charge of the C=O, then the bond multiplicity decreases, and in consequence, the vibration frequency is lower.

Two peaks at 1685 and 1570 cm^−1^ are observed in the NP spectrum (absent in cysteine spectrum). This may be produced by the vibration deformation of oxygen-containing groups [24,25]. This same vibrational deformation was observed with the signal at 3400 cm^−1^, which could correspond to the stretching vibration of O-H groups. This signal may be indicative of the interaction of copper NPs with cysteine. Weak bands observed in the FTIR spectrum at 745 cm^−1^ are assigned to C-S stretching vibrations [26]. FTIR results revealed important differences between cysteine and NPs, the S-H stretching of cysteine (2538 cm^−1^) that disappears in the NPs [25,27]. This suggests a formation of a covalent bond between the NPs and cysteine through a sulfhydryl group, a phenomenon that can be explained by the oxidation of the sulfhydryl group in the presence of metal ions. This reaction produces a Cu-SR bond that is observed at 607 cm^−1^ in the NPs FTIR spectrum [28].

Size, polydispersity, and shape of NPs were determined by TEM and DLS (Figure 3A). A monodisperse size distribution was determined for Cu_2_S NPs, with nanocrystal sizes ranging between 6 and 10 nm, as determined by DLS (Figure 3A, inset). This was also confirmed using TEM, which revealed electron-dense spherical NPs with an average size of 10 nm and mono-disperse distribution (Figure 3A).

It is well known that synthesis conditions, such as pressure, solvent, temperature, and the ratio between reagents can influence the structure of nanocrystals. Specifically, for Cu_x_S_y_ nanomaterials, there are many known stable phases (crystalline structures), ranging from copper-rich chalcocite (Cu_2_S) to copper-deficient villamaninite (CuS_2_), with other intermediate compounds in between, such as the cubic digenite, the hexagonal covellite, and the orthorhombic djurleite and anilite [29,30].

To determine the crystal structure of biomimetic NPs, HR-TEM studies were performed. HR-TEM image analysis revealed ordered atomic rows with an interplanar distance of 3.5 ± 0.1 Å (Figure 3B). The stoichiometric composition of Cu_2_S can present two different crystal symmetries: (i) the monoclinic low-chalcocite (αCh); and (ii) the hexagonal high-chalcocite (βCh). The βCh hexagonal cell appears in the P6_3_/mmc space group (No. 194) [31], and the unit cell parameters are a = 0.4005 nm and c = 0.6806 nm [32]. Considering these cell parameters, a prediction of the planes and their interplanar distances was obtained (data not shown). The planes observed in the HRTEM micrograph (Figure 3B) are separated by a distance of 3.5 ± 0.1 Å; this specific distance is predicted for the {1 0 0} family of planes. Moreover, inset of Figure 3B shows a ring diffraction pattern that fits with the predicted pattern (JEMS software prediction) for hexagonal chalcocite; in particular, the ring indexed as 1 0 0 is associated with that family of planes displaying an interplanar distance of 3.5 Å. These results indicate that biomimetic synthesis produces βCh (hexagonal) Cu_2_S nanoparticles.

Recent works have reported hexagonal phase Cu_x_S_y_ nanomaterials (pure or mixed with other structures). For example, Cu_2_S (also Cu_2_SeS and Cu_2_TeS) βCh-NPs [33], hexagonal CuS NPs [34], CuS NPs/ZnO nanorods on carbon fibers [35], and hexagonal Cu_2_S NPs [36], have been reported. Some of these reports also describe the photocatalytic activity of Cu-NPs, an important property for the application of these nanomaterials in photovoltaic technology.

Because the toxicity of Cu_x_S_y_ NPs can limit their use in some applications, we determined the biocompatibility of Cu_2_S NPs produced by our green method in cell lines and bacteria. MTS assays indicate that NPs only display a significant effect on the viability of OKF6-TERT2 cells (near 50%) at high concentrations (2000 μg/mL), but their toxicity is considerably reduced at lower concentrations, where about 95% of the cells remain viable (Figure 4). This raises a concrete possibility of the use of these Cu_2_S NPs in different biomedical applications such as modulation of cell behavior, photothermal ablation, drug delivery, and DNA detection [37].

Bacterial toxicity of biomimetic NPs was also determined in *Escherichia coli* (Figure 4B) and *Staphylococcus aureus* (Figure 4C) cells. The growth of both bacteria was not affected by treatment with biomimetic NPs at concentrations ranging from 100 to 2000 μg/mL. The absence of changes in the growth curves of *S. aureus* and *E. coli* exposed to different concentrations of NPs (Figure 4B,C) means that the viability and growth rate remain similar to control conditions without copper. 

Cu-NPs are commonly used as antibacterial materials in several applications, and their toxicity is mainly associated to Cu^2+^ release. In effect, copper oxide NPs are the most toxic metal-oxide NPs, particularly when compared with Ti, Zn, Fe, and Sb [38,39]. Chemically synthesized CuS NPs require concentrations of 10 µM CuS NPs to exert an antimicrobial effect [40]. However, in nanoparticles synthesized with amino acids, concentrations of 10 mM and longer treatment times are required to exert an antimicrobial effect [41].

For this reason, several copper oxide nanostructures are currently used in antimicrobial applications and tumor ablations [42]. Unlike CuO NPs, cytotoxicity of Cu_x_S_y_ NPs has not been studied thoroughly. CuS NPs with similar levels of biocompatibility to Au NPs, which are accepted as biocompatible nanomaterials, have been reported [43]. However, these CuS NPs were used as toxic agents for tumor ablation therapies in photothermal therapy. Therefore, the high biocompatibility of Cu_2_S NPs synthesized by our method is a novel and unexpected characteristic that favors their potential technological applications. The decreased toxicity of these Cu-NPs may be explained by high stabilization of Cu atoms inside the nanocrystal, but the detailed mechanism is currently under investigation.

### 3.2. Cu_2_S NPs Characterization as Counter Electrode

The electronic charge transport kinetics of the electrodes were analyzed by electrochemical impedance spectroscopy (EIS) using sandwiched cells (E/electrolyte/CE) [44]. FTO, Cu_2_S, and platinum were evaluated as counter electrodes. Additionally, the effect of chalcocite concentration over FTO substrate was determined. The Nyquist plots of EIS in Figure 5 showed meaningful performances of Cu_2_S/FTO (four layers) and Pt CE.

To simulate the equivalent circuit, a Randles cell was used (see Figure 5) and R_s_ and CPE values were determined [45] (Table 1). According to the circuit components, R_s_ is related to the solution resistance and CPE is a constant phase element associated to the effect of the double layer that acts as an imperfect capacitor. R_ct_ is the resistance to the electron charge transfer between the electrolyte and the counter electrode after the redox process.

The constant phase elements (CPE) were constructed by two components, a pre-exponential value associated to the magnitude of the capacitor (CPE_T_) which when compared to FTO was approximately two times higher for Cu_2_S/FTO (four layers) and three times higher for Pt CE. The exponential value (CPE_P_) was close to 1 in all cases with small differences (±0.004), indicating a capacitor-like behavior in every case.

According to the circuit components, R_s_ is related to the solution resistance, CPE is a constant phase element associated to the effect of the double layer that acts as an imperfect capacitor. R_ct_ is the resistance to the electron charge transfer between the electrolyte and the counter electrode after the redox process.

The resistance to the charge transfer (R_ct_) presents the higher difference between the cells. FTO CE showed the higher resistance value, probably due to a poisoning effect of the electrolyte solution over the electrode. On the other hand, a comparison between the number of layers of the chalcocite indicated that four layers Cu_2_S/FTO CE exhibited the lower R_CT_, confirming its excellent electrocatalytic activity (not shown). Previous works reported smaller R_ct_ values using an electrochemical deposition method of Cu_2_S nanoparticles over the surface of FTO glass [46,47].

Pt/FTO CE display the smallest R_ct_ values, almost half of those determined for Cu_2_S/FTO WE (n layers = 4). However, considering the low cost of copper and the low R_c_**_t_** values indicating the ideal performance of the electrodes, Cu_2_S NPs produced by our method constitutes an ideal material to be used as counter electrode in QDSCs.

In addition, EIS was evaluated in the presence of two electrolytes. Cu_2_S displayed a better activity for polysulfide redox reaction (data not shown), a result that is in agreement with previous reports [48] and confirm the potential of Cu_2_S NPs produced by our green method as counter electrode in QDSSCs.

To determine the performance of Cu_2_S NPs as counter electrode in QDSSC, solar cells sensitized with ruthenium dye or CdTe quantum dots were assembled. In addition, solar cells were constructed using two different electrolytes (sulphide/polysulphide or iodide/triiodide). Platinum counter electrode-based solar cells were also assessed as the control. The performance of the solar cells was tested, and photovoltaic parameters determined (Table 1).

Under one sun illumination, QDSSCs based in Cu_2_S CE presented similar parameters to those observed with the control CE Pt when two different electrolytes were tested (Table 1). Photovoltaic parameters of cells using Cu_2_S or Pt as CE were higher in QDSSCs using the high efficiency redox couple iodide/triiodide as electrolyte. In the case of sulphide/polysulphide electrolyte, a poor CE performance of Pt was found to be related to undesirable chemisorption of sulfur species on the electrolyte, a poisoning effect that has been previously reported [49]. Moreover, a slight increase in photovoltaic performance using Cu_2_S CE represents a minimum inner energy loss at electrolyte–counter electrode interfaces.

Altogether, results obtained using Cu_2_S NPs as CE on solar cells validate their performance as counter electrodes, even in the presence of different sensitizers and electrolytes. The novel characteristics of Cu_2_S NPs produced by our green method including simple synthesis, spectroscopic properties, high biocompatibility, and photovoltaic performance, validate their use in different technological applications, particularly in the generation of Cu-based QDSSCs.

## 4. Conclusions

In the present work, we reported the synthesis of Cu_2_S NPs through a simple and economical green approach in the presence of oxygen and at a low temperature. Using this one-pot protocol, we generated nanostructures exhibiting green fluorescence when irradiated with UV light, a size below 10 nm, aqueous solubility, and low toxicity in both eukaryotic cells and microorganisms. In addition, produced Cu_2_S NPs function as promising counter electrodes in photovoltaic cells with comparable performance than Pt, an expensive element widely used as CE in solar cells. Altogether, obtained results validate the use of Cu_2_S NPs produced by our method as a sustainable and economic CE alternative for energy applications.

## Figures and Tables

**Figure 1 nanomaterials-12-03194-f001:**
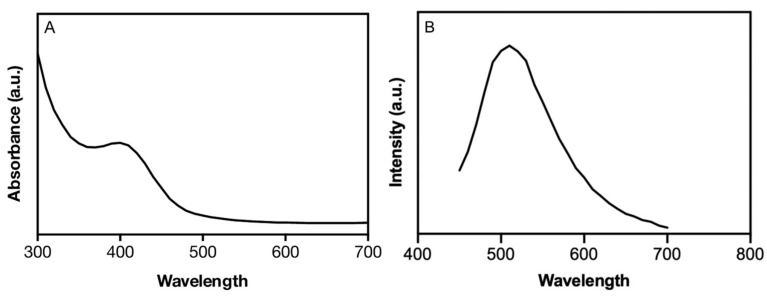
Spectroscopic properties of Cu_2_S NPs. (**A**) Absorbance and (**B**) fluorescence spectra of Cu_2_S NPs (ʎ_exc_ = 410 nm). Based on the emission spectra, a full width at half maximum (FWHM) of 103.22 nm was determined.

**Figure 2 nanomaterials-12-03194-f002:**
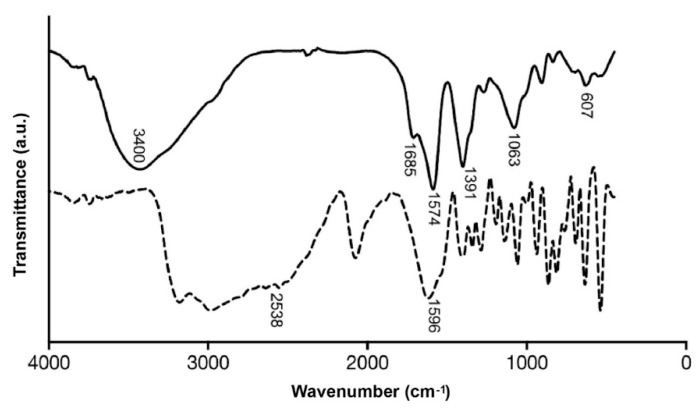
FTIR spectra of Cu_2_S NPs (complete line) and cysteine (dashed line). Interaction of the Cu metal surface with cysteine molecules was studied using FTIR spectroscopy.

**Figure 3 nanomaterials-12-03194-f003:**
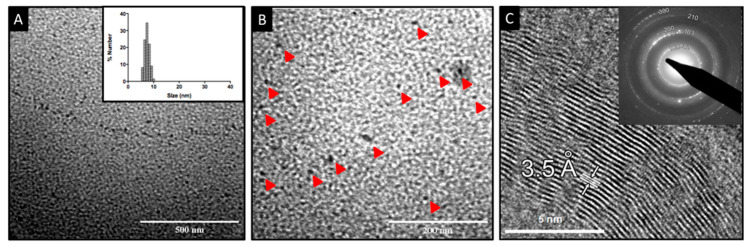
Electron microscopy characterization of NPs. (**A**) TEM image of biomimetic Cu_2_S NPs. The inset shows the size dispersion analysis through DLS. (**B**) TEM image of biomimetic Cu_2_S NPs (red arrows show the NPs). (**C**) HRTEM image obtained from the sample. A particular family of planes could be clearly distinguished from the micrograph. The corresponding interplanar distance is equal to 3.5 ± 0.1 Å. The inset in the (**C**) shows the ring diffraction pattern obtained for the sample.

**Figure 4 nanomaterials-12-03194-f004:**
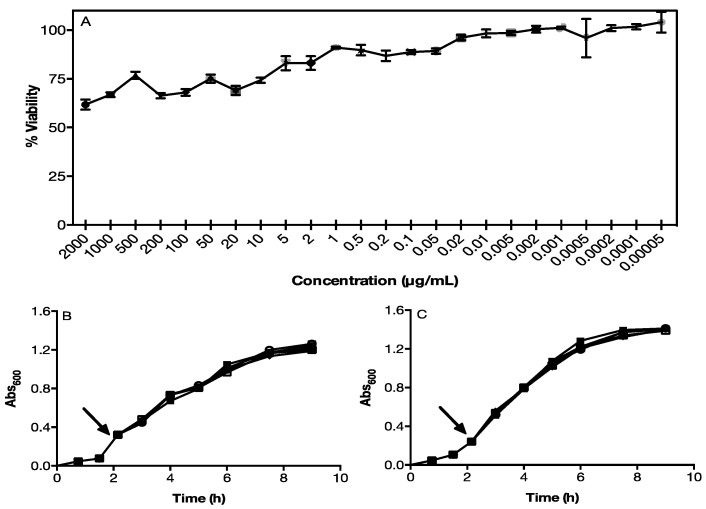
Toxicity of biomimetic NPs. (**A**) MTS viability assay on OKF6-TERT2 cells exposed to different concentrations of NPs. (**B**) Bacterial growth curves of *Staphylococcus aureus* and (**C**) *Escherichia coli* in the presence of different concentrations of NPs (the arrow indicates the time in which NPs were incorporated). Control without copper (●) 100 μg/mL (■), 500 μg/mL (○), 1.000 μg/mL (□), and 2.000 μg/mL (♦) CuCl_2_.

**Figure 5 nanomaterials-12-03194-f005:**
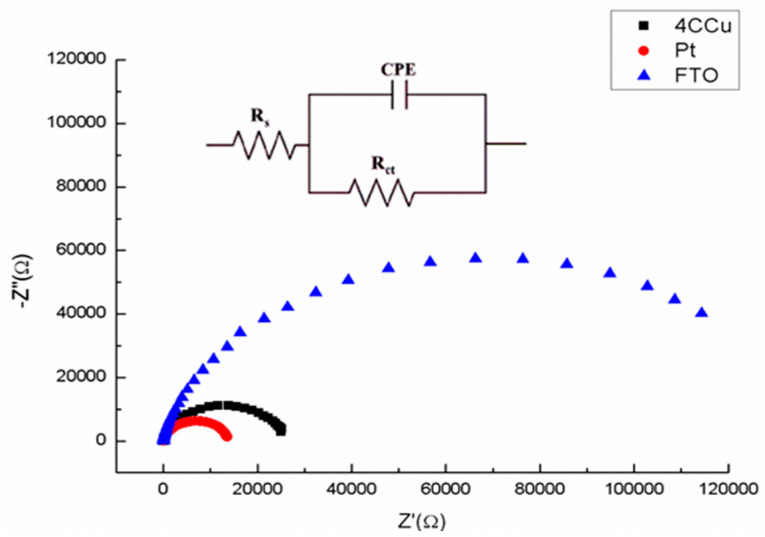
Nyquist plots obtained from EIS experiments of sandwiched cells CE/E/WE in dark conditions. FTO/TiO_2_ was used as CE, and as WE (▢) Cu_2_S/FTO modified electrode (n layers = 4), (●) Pt/FTO modified electrode and (△) FTO glass.

**Table 1 nanomaterials-12-03194-t001:** Data value of the circuit elements obtained from the fit of EIS plots of Figure 5 for each WE, and photovoltaic parameters of QDSSCs using Cu_2_S NPs or Pt as counter electrodes.

EIS Characterization							
Counter Electrode			RS/Ω		CPET/µF	CPEP	RCT/kΩ
FTO			18.15		2.82	0.94	112.7
Pt			16.81		7.83	0.93	13.8
Cu_2_S (4 layers)			26.15		5.54	0.92	25.3
**QDSSCs Characterization**							
**Sensitizer**	**Electrolyte**	**Counter Electrode**	**Short Current Density**	**Open Circuit Voltage**	**Maximum Power**	**Fill Factor**	**Efficiency**
			***J*_sc_ [A/cm^2^]**	***V*_oc_ [V]**	***P*_max_ [W]**		***ƞ* [%]**
Ruthenium	I^−^/I_3_^−^	Cu_2_S	1.50 × 10^−4^ ± 4.89 × 10^−11^	0.490 ± 4.73 × 10^−4^	1.70 × 10^−5^ ± 2.21 × 10^−12^	0.231 ± 6.67 × 10^−7^	1.70 × 10^−2^ ± 2.25 × 10^−6^
		Pt	5.20 × 10^−4^ ± 1.04 × 10^−10^	0.583 ± 3.02 × 10^−4^	1.92 × 10^−4^ ± 6.75 × 10^−11^	0.632 ± 4.47 × 10^−5^	1.92 × 10^−1^ ± 6.78 × 10^−5^
CdTe QDs	S_n_^2−^/S^2^	Cu_2_S	5.02 × 10^−5^ ± 1.21 × 10^−10^	0.300 ± 5.15 × 10^−4^	5.40 × 10^−6^ ± 2.09 × 10^−12^	0.358 ± 5.29 × 10^−4^	5.40 × 10^−3^ ± 2.09 × 10^−6^
		Pt	4.49 × 10^−5^ ± 1.07 × 10^−10^	0.207 ± 6.47 × 10^−5^	4.04 × 10^−6^ ± 8.00 × 10^−13^	0.436 ± 4.85 × 10^−4^	4.04 × 10^−3^ ± 7.84 × 10^−7^

## Data Availability

The data is available on reasonable request from the corresponding author.

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
