# Peer review of "Cysteine-Mediated Green Synthesis of Copper Sulphide Nanoparticles: Biocompatibility Studies and Characterization as Counter Electrodes"

_nanomaterials, 2022, doi:10.3390/nano12183194_

Round 1
Reviewer 1 Report
This paper presents a synthesis method for Cu2S NPs by simply heating an aqueous solution of CuCl2, L-cysteine and borax-citrate buffer to 37°C for 12h. The obtained 6-10 nm Cu2S NPs were characterized using TEM, DLS, UV-vis, PL and FTIR spectroscopy. Their cytotoxicity and potential use as counter electrode in DSSCs was also evaluated.
Even though this paper contains some points of interest, the results appear too preliminary for publication. In particular, the synthesis reaction is not at all discussed and only one set of experimental parameters is presented yielding only one size of NPs. Moreover, the composition of the obtained NPs remains elusive as no elemental analysis data is presented. However, the Cu:S ratio is an important parameter, in particular when discussing the eventual SPR properties of the NPs. Finally, the proof-of-concept for the proposed main application of the Cu2S NPs, i.e., their use as counter electrode in DSSCs, suffers from the very low efficiency of the solar cells in this study.
The link between the use of Cu2S as counter electrode in DSSCs and their fluorescence is not clear in the introduction.
The abstract states that the reaction was carried out at RT (i.e. 25°C), however, in the Experimental Section it is written that 37°C have been used.
The PL data (Fig. 1b) looks truncated (base line is missing). Furthermore, the PLQY should be determined (e.g., by comparison with an appropriate dye) and the PL linewidth should be indicated.
The discussion of the SPR (p. 4) is inappropriate: if the material is fluorescent, it cannot exhibit at the same time a surface plasmon resonance.
To identify wether the peak at 410 nm observed in Fig. 1a stems from excitonic absorption features, different sized particles should be prepared.
In the discussion of the FTIR spectrum (p. 5) the position of the carbonyl signal of the carboxylic acid function depends on whether it is protonated or not and on the binding mode with the NP surface. ‘Wavelength’ needs to be replaced by ‘Wavenumber’
TEM (Fig. 3): a higher magnification image would be needed that that given in panel A to appreciate the size and size distribution. Panel B: the analyzed structure does not seem to be related to a Cu2S nanoparticle but rather to some kind of side-product, it looks not spherical.
Toxicity: In Fig. 4 A on can see that the cell viability is reduced by around 25% for low concentrations of Cu2S NPs (5-10 microgram/mL). The authors conclude that at lower concentrations than 2000 microgram/mL the NPs were ‘practically harmless’, which sounds vague and incorrect. Also in the next phrase they compare with ppm concentrations and it is not clear how these values can be translated to the present study.
PV: on a relative scale, an improvement over the performance of DSSCs and QDSSCs using Pt counterelectrodes is observed when using Cu2S NPs instead. However, on an absolute scale, the PCE of all devices is that low (0.008-0.023%) that it becomes difficult to draw any reliable conclusions, other factors could mask the influence of the counter electrode. The fill factor values and J/V curves should be given. What's about the stability of the solar cells? if the J/V sweep is run again, is the same curve obtained?
Reviewer 2 Report
The authors reported a one-pot green method for aqueous synthesis of fluorescent copper sulphide nanoparticles (NPs), and studied the biocompatibility and characterization as counter electrodes. The writing is well. However, some revisions are needed before acceptation.
1. In the introduction, the novelty is not clear. Since CuxS NPs have been reported in various study, as cited by the authors in reference [12-17], yet what is the improvement made by the authors? What the difference between the NPs prepared by the authors and other research? The authors should give clear comparisons and comments with other existed CuxS NPs besides the preparation. The only improvement of sample preparation just isn't enough for publication in such journal.
Moreover, there are too many paragraphs in the introduction. Some should be merged.
2. It is strange to present two different research orientation (biology and electrochemistry) in one paper. The focal points of the paper are not prominent.
3. Figure 4. Lack of interpretation of figure 4c in the legend.
Round 2
Reviewer 1 Report
The authors addressed many of the points I raised, but I don't understand why the FF and J/V curves have still not been integrated as these data must have been recorded during the PV characterization.
In the present form, the Supp. Information is superfluous, the data could simply be added to Table 1 in the main manuscript.
Author Response
Response: Done. We modified the ms to include the suggestion of reviewer 1. As requested, the information of Supplementary Figure 1 was incorporated in Table I in the new version of the ms.
Regarding the FF and J/V curves, as indicated by the reviewer, this data was obtained during the photovoltaic studies and used to determine the parameters included in Table I. However, we decided not to include them in the ms to focus on the most significant aspects of this research: the synthesis method, the characterization of the NPs, and their use as CE (Nyquist plots).